# Collective excitation of plasmon-coupled Au-nanochain boosts photocatalytic hydrogen evolution of semiconductor

Guiyang Yu[1], Jun Qian[2], Peng Zhang [3], Bo Zhang[1], Wenxiang Zhang[1], Wenfu Yan [1] & Gang Liu [1]*

Localized surface plasmon resonance (LSPR) offers a valuable opportunity to improve the efficiency of photocatalysts. However, plasmonic enhancement of photoconversion is still limited, as most of metal-semiconductor building blocks depend on LSPR contribution of isolated metal nanoparticles. In this contribution, the concept of collective excitation of embedded metal nanoparticles is demonstrated as an effective strategy to enhance the utilization of plasmonic energy. The contribution of Au-nanochain to the enhancement of photoconversion is 3.5 times increase in comparison with that of conventional isolated Au nanoparticles. Experimental characterization and theoretical simulation show that strongly coupled plasmonic nanostructure of Au-nanochain give rise to highly intensive electromagnetic field. The enhanced strength of electromagnetic field essentially boosts the formation rate of electron-hole pair in semiconductor, and ultimately improves photocatalytic hydrogen evolution activity of semiconductor photocatalysts. The concept of embedded coupled-metal nanostructure represents a promising strategy for the rational design of high-performance photocatalysts.

[1] State Key Laboratory of Inorganic Synthesis and Preparative Chemistry, College of Chemistry, Jilin University, 2699 Qianjin Road, 130012 Changchun, China. [2] School of Physics, Nankai University, 300071 Tianjin, China. [3] Department of Chemistry, Dalhousie University, 6274 Coburg Road, Halifax B3H4R2, Canada. *email: lgang@jlu.edu.cn

Enhancing the efficiency of semiconductor photocatalysts is of paramount importance for realizing more efficient conversion of solar energy in artificial photosynthesis[1–7]. Integrating a plasmonic metal nanostructure with a semiconductor has been found as a promising alternative to improve the efficiency of conventional architectures[8–11]. The enhanced photoreactivity is attributed to the localized surface plasmon resonances (LSPRs) of plasmonic metal, in which confined free electrons oscillate with the same frequency as the incident radiation, giving rise to intense, highly localized electromagnetic fields[12–16]. However, plasmonic enhancement of photoconversion is still limited, far from reaching the theoretical maximum efficiency of plasmonic metal–semiconductor[17]. To advance this emerging method, some basic problems, such as the architectures and fabrication strategies of plasmon building blocks, need to be carefully reconsidered.

Three major energy transfer mechanisms between plasmonic metals and semiconductors have been proposed in the past decade: light scattering, hot electron injection, and plasmon-induced resonance energy transfer[10,18–20]. A big challenge is how to design a plasmonic metal–semiconductor heterostructure for offering great possibility in enabling above three major plasmonic energy transfer mechanisms. Primarily, the proximity of semiconductor to plasmonic metal should be one of important factors to improve the energy transfer. It is not only because the plasmonic hot electrons need an interfacial transfer through overcoming a Schottky barrier, but also the distance could significantly lower the plasmon-induced resonance energy transfer from the metal to the semiconductor[12,21–23]. The localized electromagnetic fields of plasmonic metal are spatially non-homogenous. The highest intensity is at the surface of the plasmonic metal and decreasing exponentially distance from the surface[24–26]. Much closer to the plasmonic metal, more electromagnetic fields can be captured. However, most of current researches based on a nanostructure with isolated metal particles dispersed on semiconductor surface. The proximity of semiconductor to the electromagnetic fields of plasmonic metal is limited for such architectures.

In addition, enhancing the local intensity of plasmon-induced electromagnetic fields is another crucial factor for maximizing efficiency of plasmonic metal–semiconductor. We have known that it partly depends on the competence of particle nature, including the composition, size, and shape[27–31]. While the collective behaviors of dense metal nanoparticles to the plasmonic enhancement is still unclear. Although some physical research show that strongly coupled metallic nanostructures would generate much higher electromagnetic fields at the adjacent spot, what kind of architectures would be effective for the semiconductor photocatalyst systems is still unknown[32–34]. It should be simultaneously considered the spatial arrangement of semiconductor and plasmonic metals. It would significantly depend on a rational design of plasmonic metal–semiconductor building blocks.

In this contribution, considering both the strength and transfer of plasmonic energy, a plasmonic building block that embedding a strongly plasmon-coupled metal nanostructure is fabricated. Au-nanochain (a strongly plasmon-coupled metal nanostructure) is built into $Zn_xCd_{1-x}S$ semiconductor (denoted as Au-chain@$Zn_xCd_{1-x}S$, $x = 0.67$). Visible-light-driven hydrogen evolution rate of $Zn_{0.67}Cd_{0.33}S$ can be boosted to 16,420 $\mu mol\,h^{-1}\,g^{-1}$ by the Au-nanochain. The apparent quantum efficiency can reach 54.6% at 420 nm. All these results are obtained without additional cocatalysts. The contribution of Au-nanochain to the enhancement of photoconversion is 3.5 times increase in comparison with that of conventional isolated Au nanoparticles. A systematic investigation is carried out to clarify the dependence of the photocatalytic performance on the spatial arrangement of plasmonic

metal in the metal–semiconductor building blocks. Both the theoretical simulation and experimental characterization shows that Au-nanochain could give rise to much higher local electromagnetic field than that of isolated Au nanoparticles when excited by the incident light. It essentially increases the formation of the electron–hole pair on the nearby semiconductors and causes a high reactivity. The concept of embedding coupled-metal nanostructure in semiconductors represents an efficient way to enhance the solar energy conversion efficiency.

## Results

**Preparation and structural analysis of Au-nanochain-containing sample.** Au-chain@$Zn_xCd_{1-x}S$ ($x = 0.67$) was prepared by a hydrothermal method. Figure 1a illustrates the preparation procedures of the heterostructure from ions. Firstly, L-cysteine solution was mixed with $Zn(NO_3)_2$ or $Cd(NO_3)_2$ to form the stable complexes of cysteine-$Zn^{2+}$/$Cd^{2+}$. Next, Au colloids with uniform particle size of 15 nm was added to the freshly prepared cysteine-$Zn^{2+}$/$Cd^{2+}$ solution under vigorous stirring. Finally, the cysteine-$Zn^{2+}$/$Cd^{2+}$-coupled Au colloids were transferred into Teflon-lined stainless-steel autoclaves and maintained at 130 °C for 6 h. The Au colloids were prepared by reducing $HAuCl_4$ with sodium citrate in an aqueous solution (see details in the "Methods" section and Supplementary Figs. 1 and 2). The measured elemental composition of Zn/Cd from inductively coupled plasma (ICP) spectroscopy is closely to the concentration of $Zn^{2+}$/$Cd^{2+}$ used during the synthesis. Transmission electron microscopy (TEM) image shows the spatial arrangement of Au nanoparticles in Au-chain@$Zn_{0.67}Cd_{0.33}S$ (Fig. 1b). Au nanoparticles connect with each other forming a chain shape. Some branches can be observed in the image, but all the Au nanoparticles are closely packed. $Zn_{0.67}Cd_{0.33}S$ grows around the Au-nanochains and wrapped them completely. No porous structure can be found in the images. High-resolution transmission electron microscopy (HRTEM) image (Fig. 1c), clearly shows the fringe of 0.24 nm, which is ascribed to the face-centered cubic (fcc) Au (111). While another set of lattices with periodic spacing of 0.31 nm are assigned to cubic $Zn_{0.67}Cd_{0.33}S$ (101) lattice planes. Two sets of reflections can be observed in selected area electron diffraction (SAED) patterns (Fig. 1b inset). High-angle annular dark-field (HAADF) image and energy-dispersive X-ray spectroscopy (EDXS) mapping images (Fig. 1d), could also clearly depict the arrangement of Au nanoparticles, while Zn, Cd, and S are homogeneously distributed on Au-chain@$Zn_{0.67}Cd_{0.33}S$.

**Theoretical simulation.** Three-dimensional finite difference time domain (FDTD) method is used to simulate the near-field distributions of isolated and coupled nanoparticles (Fig. 1e and Supplementary Figs. 3–5). The total field scattered field (TF/SF) source is used in simulation. The incident wavelength is 520 nm. The grid spacing is 0.5 nm. The radius of gold nanosphere is 7.5 nm. Figure 1e shows the electric field distributions of Au nanoparticles with different interparticle distances. When distance decrease from 20 to 2 nm, the strength of electromagnetic fields increase at least one order of magnitudes at hot spot area. It demonstrates that Au-nanochain can give rise to highly intense and localized electromagnetic fields when excited by incident light of the appropriate polarization.

**Photocatalytic performance.** Photocatalytic test of Au-chain@$Zn_{0.67}Cd_{0.33}S$ under visible light (≥420 nm) gives an activity of 16,420 $\mu mol\,h^{-1}\,g^{-1}$ for hydrogen evolution (Fig. 1d). It is about 20 times of commercial CdS (830 $\mu mol\,h^{-1}\,g^{-1}$), and 3.3 times of pure $Zn_{0.67}Cd_{0.33}S$ (5020 $\mu mol\,h^{-1}\,g^{-1}$) prepared with the same method as Au-chain@$Zn_{0.67}Cd_{0.33}S$. It should be

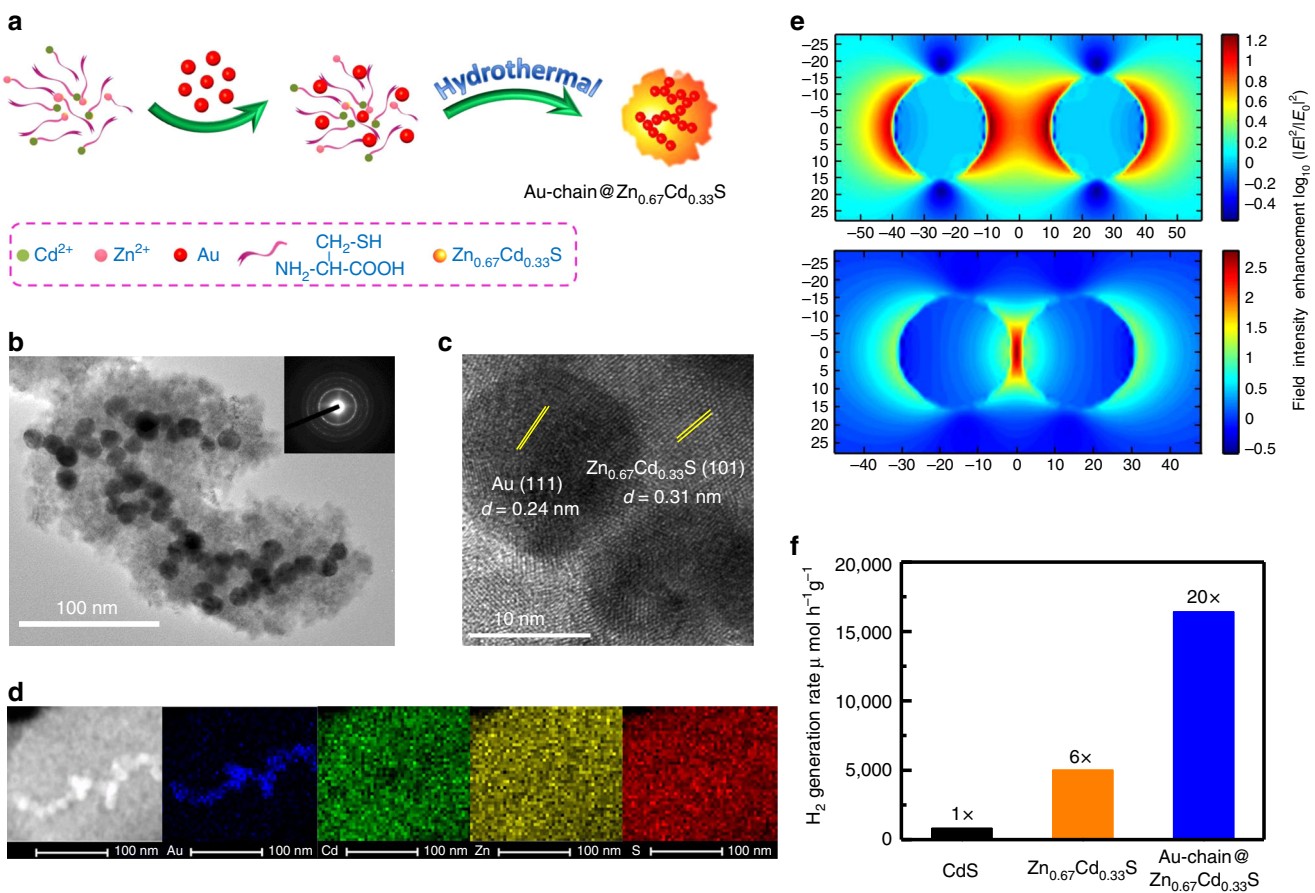

**Fig. 1** Synthesis and physicochemical properties of photocatalysts. **a** Schematic illustration of the preparation process of Au-chain@$Zn_{0.67}Cd_{0.33}S$. **b** TEM and SAED patterns (inset), **c** HRTEM image, and **d** HAADF and EDXS mapping images of Au-chain@$Zn_{0.67}Cd_{0.33}S$. **e** FDTD simulation of the near-field distributions of isolated and coupled nanoparticles excited by visible light. **f** Photocatalytic $H_2$ evolution activity of commercial CdS, pure $Zn_{0.67}Cd_{0.33}S$ and Au-chain@$Zn_{0.67}Cd_{0.33}S$ under visible light irradiation (≥420 nm)

noted that all these catalytic results are obtained in the absence of any additional metal or metal oxide as a cocatalyst on the surface of semiconductor. Under the optimized reaction conditions, the apparent quantum efficiency of Au-chain@$Zn_{0.67}Cd_{0.33}S$ can even reach to 54.6% under 420 nm illumination (Supplementary Fig. 6). This efficiency is obviously higher than most of literature results, and keeps leading in the tests free of cocatalysts (Supplementary Table 1). The high efficiency should attribute to the strongly coupled nanostructure of Au-nanochain embedded into the semiconductor.

**Influence of the spatial arrangement of plasmonic metals.** For systematic investigation of the dependence of the photocatalytic performance on the spatial arrangement of plasmonic Au, two other samples Au-iso@$Zn_{0.67}Cd_{0.33}S$ (Au nanoparticles separately embedded into $Zn_{0.67}Cd_{0.33}S$, see Supplementary method and Supplementary Figs. 7 and 8) and Au-surf@$Zn_{0.67}Cd_{0.33}S$ (Au nanoparticles post loaded on the surface of $Zn_{0.67}Cd_{0.33}S$, Supplementary method and Supplementary Figs. 7 and 9) were also prepared. Au-iso@$Zn_{0.67}Cd_{0.33}S$ and pure $Zn_{0.67}Cd_{0.33}S$ were prepared with a hydrothermal method, which is the same as Au-chain@$Zn_{0.67}Cd_{0.33}S$. Figure 2a shows the X-ray diffraction (XRD) patterns of Au-chain@$Zn_{0.67}Cd_{0.33}S$, Au-iso@$Zn_{0.67}Cd_{0.33}S$, Au-surf@$Zn_{0.67}Cd_{0.33}S$ as well as pure $Zn_{0.67}Cd_{0.33}S$. All the samples exhibit six diffraction peaks at $2\theta = 25.1°$, 26.8°, 28.4°, 44.0°, 48.1°, and 52.1°, which can be assigned to (100), (002), (101), (110), (103), and (112) planes of hexagonal wurtzite $Zn_{0.67}Cd_{0.33}S$ (JCPDS No. 40-0835)[35]. The

diffraction peaks of Au nanoparticles cannot be detected in the XRD patterns, due to the low concentration and the highly dispersed state of Au nanoparticles in the samples. $N_2$-adsorption results (Table 1) show that the specific surface areas of three Au-containing samples are all at about 40 m$^2$ g$^{-1}$, which is just a little higher than that of pure $Zn_{0.67}Cd_{0.33}S$ (36.1 m$^2$ g$^{-1}$). The isotherms (Supplementary Fig. 10) shows that all these samples are non-porous materials, which is consistent with that of TEM results.

Figure 2b, c shows the X-ray photoelectron spectroscopy (XPS) of Au-chain@$Zn_{0.67}Cd_{0.33}S$, Au-iso@$Zn_{0.67}Cd_{0.33}S$, Au-surf@$Zn_{0.67}Cd_{0.33}S$, and pure $Zn_{0.67}Cd_{0.33}S$. All these four samples exhibit similar surface chemical state of Cd and Zn. The surface Zn/Cd ratios of these samples are all close to 2.0 (Table 1), which is nearly the same as the composition of the bulk. It should be noted that no signal of Au can be detected on the surface of Au-chain@$Zn_{0.67}Cd_{0.33}S$, Au-iso@$Zn_{0.67}Cd_{0.33}S$ (Table 1 and Supplementary Fig. 11), confirming that Au nanoparticles in these two samples are embedded completely into the $Zn_{0.67}Cd_{0.33}S$ semiconductor.

Figure 3a shows the Raman spectroscopy of Au-chain@$Zn_{0.67}Cd_{0.33}S$, Au-iso@$Zn_{0.67}Cd_{0.33}S$, Au-surf@$Zn_{0.67}Cd_{0.33}S$, and pure $Zn_{0.67}Cd_{0.33}S$. Two typical Raman scattering peaks of wurtzite $Zn_{0.67}Cd_{0.33}S$ can be observed in the spectroscopy, which are the first (1LO) and second (2LO) band of longitudinal optical (LO) phonon modes at 295 and 598 cm$^{-1}$, respectively[36]. The intensity of Raman signals follows the order of Au-chain@$Zn_{0.67}Cd_{0.33}S$ > Au-iso@$Zn_{0.67}Cd_{0.33}S$ > Au-surf@$Zn_{0.67}Cd_{0.33}S$ > $Zn_{0.67}Cd_{0.33}S$.

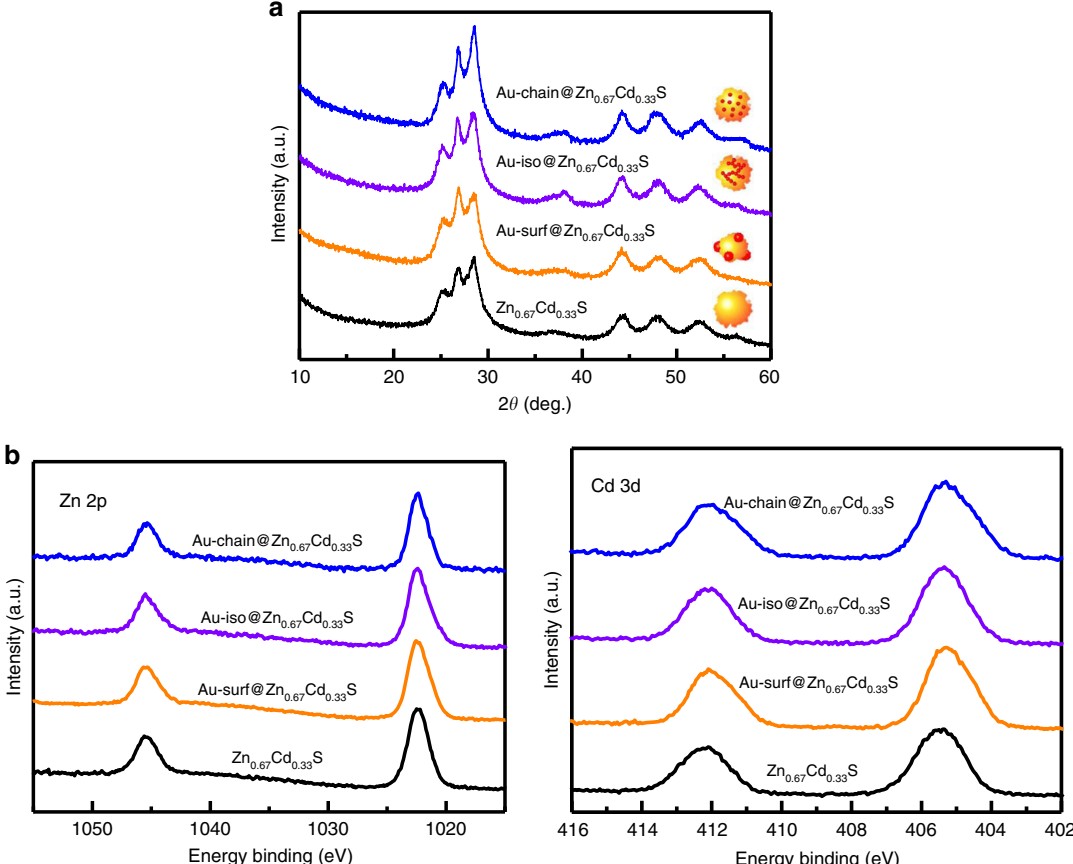

**Fig. 2** Crystal structure and surface properties. **a** XRD patterns, **b** Zn 2p and Cd 3d XPS spectra of pure $Zn_{0.67}Cd_{0.33}S$ and $Au@Zn_{0.67}Cd_{0.33}S$ with different spatial arrangement of Au nanoparticles

| Table 1 Texture properties and the atom contents on the surface of various samples | | | | | |
|---|---|---|---|---|---|
| **Samples** | $S_{BET}$ (m$^2$ g$^{-1}$) | **XPS spectra results** | | | |
| | | Zn (at.%) | Cd (at.%) | Au (at.%) | Zn:Cd |
| $Zn_{0.67}Cd_{0.33}S$ | 36.1 | 30.89 | 15.45 | 0 | 2.00:1 |
| $Au$-$surf@Zn_{0.67}Cd_{0.33}S$ | 40.1 | 30.47 | 14.87 | 0.14 | 2.05:1 |
| $Au$-$iso@Zn_{0.67}Cd_{0.33}S$ | 41.3 | 31.26 | 15.72 | 0 | 1.99:1 |
| $Au$-$chain@Zn_{0.67}Cd_{0.33}S$ | 40.9 | 31.72 | 15.81 | 0 | 2.01:1 |

According to the principle of surface-enhanced Raman spectroscopy (SERS), the enhancement of signals relies on the received plasmon-induced local electromagnetic field from metal nanostructures[37]. XRD and XPS have shown that $Zn_{0.67}Cd_{0.33}S$ in all these samples possess nearly the same structure and surface properties. So, compared with pure $Zn_{0.67}Cd_{0.33}S$, the enhancement of Raman signals in Au-containing samples should be ascribed to the plasmon-induced local electromagnetic field of Au nanostructures. The field received by $Zn_{0.67}Cd_{0.33}S$ follows the order of embedded Au nanochain > embedded Au nanoparticles > surface Au nanoparticles.

UV–vis spectra further confirm above results (Fig. 3b). In comparison with pure $Zn_{0.67}Cd_{0.33}S$, Au-containing samples show a red shift of absorption edge, which are due to the influence of LSPRs from Au nanostructure. This shift originates from the change of optical properties of $Zn_{0.67}Cd_{0.33}S$ affected by the electromagnetic effect of Au LSPR, while not a simple addition of absorption of $Zn_{0.67}Cd_{0.33}S$ and Au LSPR (Supplementary Figs. 1 and 12). Figure 3b inset shows the difference spectra for the composite samples, which is obtained by substracting the $Zn_{0.67}Cd_{0.33}S$ spectrum from the composite spectra. Obviously, Au nanochains exhibit the greatest contribution to the increase of the optical absorption of the composite catalyst (Au-chain@$Zn_{0.67}Cd_{0.33}S$) in the range of 430–480 nm, followed by embedded Au nanoparticles (Au-iso@$Zn_{0.67}Cd_{0.33}S$) and the ones on the surface (Au-surf@$Zn_{0.67}Cd_{0.33}S$).

Both Raman and UV–vis results provide the experimental evidences that embedded Au-nanochain has greatest impact on the excitation of $Zn_{0.67}Cd_{0.33}S$. It should originate from the electromagnetic properties of interacting Au nanoparticles in close mutual proximity. In brief, the interaction energy can be described with $V \propto p_1p_2/r^3$, where $p_1$ and $p_2$ are the magnitudes of the diple moments and $r$ is the inter particle distance[38]. This interaction energy is considerably stronger in the case of nearly adjacent Au nanoparticles[39]. FDTD simulations shown in Figs. 1c and 3c further match the change of spectroscopy observed experimentally. The adjacent nanoparticle pairs showing a hot spot in the junction for incident polarization along the interparticle axis, where the strength of electromagnetic field at least one order of magnitudes in comparison with that of isolated particles.

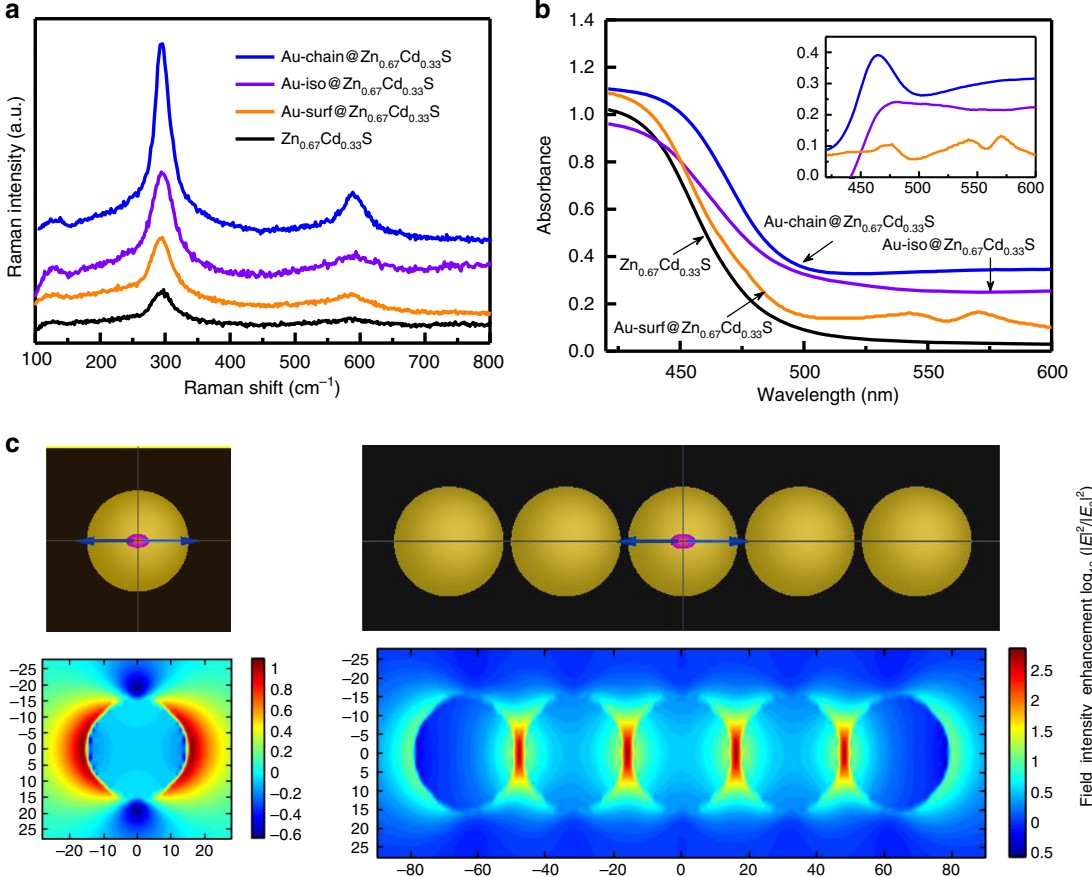

**Fig. 3** Plasmon-induced local electromagnetic field effect. **a** Raman spectra. **b** UV-vis diffuses reflection spectra of pure $Zn_{0.67}Cd_{0.33}S$ and $Au@Zn_{0.67}Cd_{0.33}S$ with different spatial arrangement of Au nanoparticles. **c** FDTD simulation of the near-field distributions of single Au nanoparticle and five coupled Au nanoparticles

The enhancement of photocatalytic hydrogen evolution activity depended on the spatial arrangement of plasmonic Au nanostructure is shown in Fig. 4a. Compared with pure $Zn_{0.67}Cd_{0.33}S$ (5020 $\mu mol\,h^{-1}\,g^{-1}$), Au-surf@$Zn_{0.67}Cd_{0.33}S$ and Au-iso@$Zn_{0.67}Cd_{0.33}S$ enhance this activity to 8290 and 11,560 $\mu mol\,h^{-1}\,g^{-1}$, respectively. The difference of these two samples is that isolated Au nanoparticles loaded on the surface for Au-surf@$Zn_{0.67}Cd_{0.33}S$ and embedded in the body for Au-iso@$Zn_{0.67}Cd_{0.33}S$. The different enhancement of photocatalytic activity is mainly due to the the proximity of $Zn_{0.67}Cd_{0.33}S$ to the electromagnetic field of plasmonic Au (Fig. 4b). Embedded structure facilitates the transfer of plasmonic energy from Au nanoparticle to the nearby $Zn_{0.67}Cd_{0.33}S$ semiconductor. All these activities are much lower than that of Au-chain@$Zn_{0.67}Cd_{0.33}S$ (16420 $\mu mol\,h^{-1}\,g^{-1}$). The contribution of Au-nanochain to the enhancement of photoconversion is about 3.5 times increase in comparison with that of isolated Au nanoparticles. Under the similar reaction conditions, the apparent quantum yield is 14.2%, 19.3%, 28.7%, and 43.5% for these four samples, respectively (Supplementary Fig. 6). The enhancements of these apparent quantum yields are consistent with the $H_2$ evolution rate. The apparent quantum yield could be further optimized to 54.6% when increasing the amount of catalyst. The plasmon-coupled Au nanochain in Au-chain@$Zn_{0.67}Cd_{0.33}S$ should play a crucial role for the enhancement of activity. In addition, it was observed that Au-chain@$Zn_{0.67}Cd_{0.33}S$ is also with a high stability in the reaction process (Supplementary Fig. 13). No changes were measured both in crystal structure and surface properties (XRD patterns and XPS spectra, Supplementary Figs. 14 and 15). It

indicates that the increase of electromagnetic fields do not lower the stability of the semiconductor. All these evidences confirm that embedded Au-nanochain offers a opportunity to improve the photocatalytic performance of semiconductor photocatalysts via the increased electromagnetic energy field.

The intensity and proximity of electromagnetic fields essentially affect the rate of $e^-/h^+$ formation in the $Zn_{0.67}Cd_{0.33}S$ semiconductor. Figure 4c shows the photoluminescence (PL) spectra of above four photocatalysts. It is known that photo-excited electron and hole recombine each other through several processes, such as direct band-to-band coupling and/or shallowly/deeply trapped potential states[40–42]. The intensity of the spectra can reflect the amount of electrons recombined with holes under emission of photons. It can be observed that Au nanochain significantly improves the emission signal of $Zn_{0.67}Cd_{0.33}S$, followed by isolated Au nanoparticles. Time-resolved PL measurements (Fig. 4d) show that the intensity of Au-chain@$Zn_{0.67}Cd_{0.33}S$ decays much more slowly than that of other composite photocatalysts, and pure $Zn_{0.67}Cd_{0.33}S$, indicating a longer lifetime of electron–hole pairs in Au-chain@$Zn_{0.67}Cd_{0.33}S$. This longer lifetime should be correlated with both the increasing amount of electron–hole pairs and the presence of surface-trapped states. Fitting results of Cd $3d$ XPS spectra show that there are certain amount of Cd species with low valent states on the surface of these four samples (Supplementary Fig. 16). Our previous work have shown that these Cd species act as the trap sites for photo-excited electrons and active sites for hydrogen evolution[40,41]. In this case, a broad emission signal in the range of 450–650 nm can be observed. No obvious shift among the signals

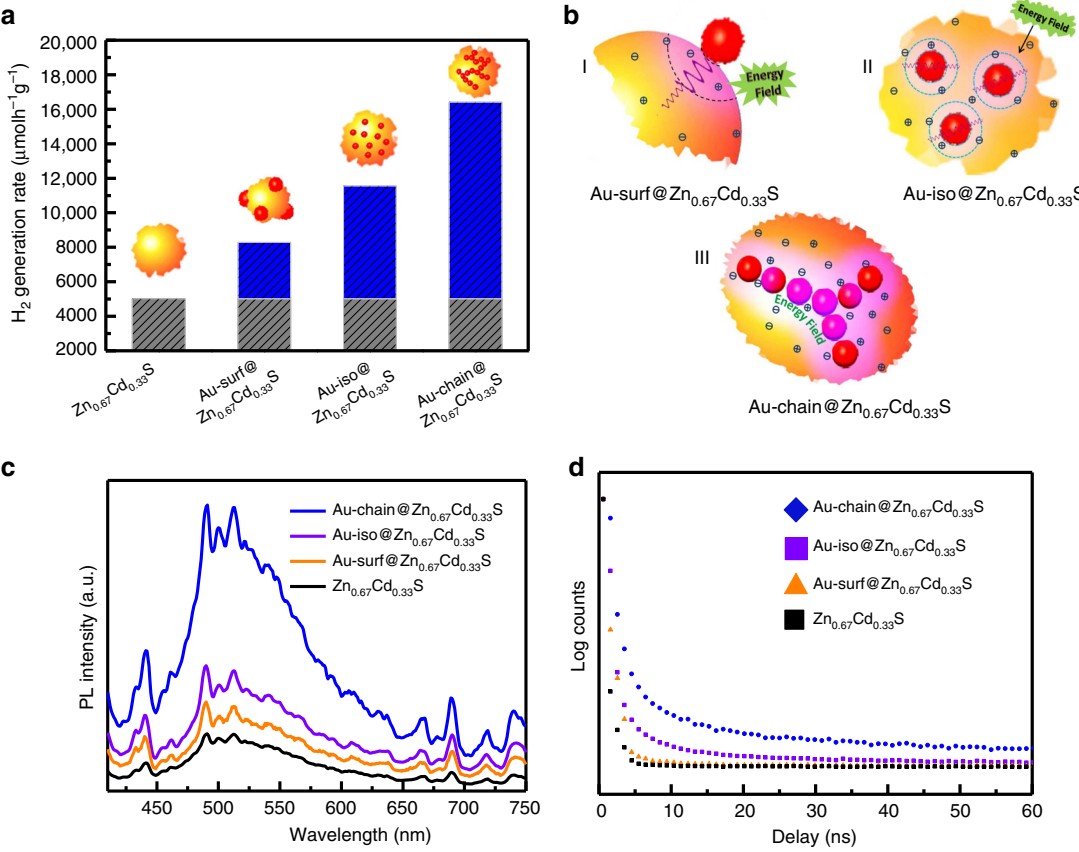

**Fig. 4** Photocatalytic performance and mechanistic insights. **a** Photocatalytic $H_2$ evolution activity of pure $Zn_{0.67}Cd_{0.33}S$ and $Au@Zn_{0.67}Cd_{0.33}S$ with different spatial arrangement of Au nanoparticles under visible light irradiation (≥420 nm); blue columns represent the enhancement promoted by plasmonic-Au nanostructures. **b** Illustration for the proximity of semiconductor to the electromagnetic fields of plasmonic Au nanoparticles. **c** Photoluminescence (PL) emission spectra of pure $Zn_{0.67}Cd_{0.33}S$ and different $Au@Zn_{0.67}Cd_{0.33}S$ samples, excitation wavelength: 390 nm. **d** PL lifetime decay of pure $Zn_{0.67}Cd_{0.33}S$ and different $Au@Zn_{0.67}Cd_{0.33}S$ samples, excitation wavelength: 400.8 nm

of these samples should be mainly due to the similar crystal structure and surface properties of $Zn_{0.67}Cd_{0.33}S$ semiconductor in these four samples. This broad emission peak indicates the presence of multiple radiation processes of excited electrons, including the emission from the band edge and the surface-trapped states. These trapped sites could play a more important role in sample of Au-chain@$Zn_{0.67}Cd_{0.33}S$. With the increase of electron–hole pairs formation, much more electrons could be trapped by these sites and effectively extend the lifetime of electron–hole pairs.

**Discussion**

The Au nanoparticles in Au-iso@$Zn_{0.67}Cd_{0.33}S$ and Au-chain@$Zn_{0.67}Cd_{0.33}S$ are wrapped well by $Zn_{0.67}Cd_{0.33}S$ semiconductor. Little porous structure can be detected in these samples. So, the role of Au nanoparticles as a cocatalyst can be excluded. All the enhancements can be ascribed to the plasmonic effect of Au nanoparticles in the metal–semiconductor building blocks. In addition, the Au nanoparticles are protected by citric acid groups in the colloids aqueous solution and coupled with *L*-cysteine during the hydrothermal synthesis. There should be a non-conductive organic layer between Au nanoparticle and sulfide compound. It can be preliminarily verified by FT-IR measurement (Supplementary Fig. 17). This structure is suitable for the energy transfer from plasmonic Au to the semiconductor and could avoid the quenching of charge carriers on the surface of Au nanoparticles.

Combining all the above results, it can be concluded that the proximity and the spatial arrangement of plasmonic Au are two important factors affecting the photocatalytic performance. Firstly, the structure of embedding Au into the body of semiconductor effectively improves the proximity, facilitating the transfer and maximizing the use of electromagnetic energy from plasmonic Au to semiconductor (Fig. 4b). As for the isolated Au nanoparticle, theoretical simulation reported by Thomann et al. over 50 nm Au nanoparticles showed that the strength of electromagnetic field vary little whether the nanoparticle locates on the top or the bottom of the semiconductor[43]. So, shorter distance between the semiconductor and plasmonic Au could realize more energy transfer. Secondly, the strongly coupled nanostructure of Au nanochain shows stronger capability of improving local electromagnetic energy field. The collective excitation of plasmonic metal facilitate much more plasmonic energy transfer from metal to the semiconducor. This energy enhances the formation rate and lifetime of $e^-/h^+$ pairs. Combined with quantum efficiency and stability measurements, it can further confirm that the coupled metal nanoparticles lead to an increase in lifetime of photo-excited electron–hole pairs. The significant enhancement of photocatalytic activity should origin from the collective behavior of these coupled metal nanoparticles, which is quite different from simple plasmonic effect of the isolated one. All these results show that embedding the plasmon-coupled metal nanostructure into semiconductor would be an effective strategy to fully take advantage of plasmonic energy.

It is known that Au nanochain is only one simple nanostructure with plasmon-enhancement effect. Considering a great deal of coupled metallic nanostructures has been reported in physics and nanoscience area, embedding different kinds of coupled nanostructure into semiconductor could further optimize the efficiency of photocatalysts. In addition, recombination of photoexcited electron–hole pairs is present in all semiconductor materials. Considering this recombination, the Au nanoparticle closer to the surface could favorite more charge carriers been utilized. Therefore, there should be an optimized depth of Au nanoparticle due to the balance of energy utilization (electron–hole pair formation) and charge carriers recombination. It correlates with both the nature of semiconductors and their crystallizations. Further work is still needed on the investigation of various kinds of coupled plasmonic metal nanostructure and optimizing the placement of these metal nanoparticles in the semiconductors.

In summary, Au-nanochain with strongly coupled plasmonic nanostructure was demonstrated to be an effective building block to enhance the photocatalytic performance of nearby semiconductor. The chain structure possesses junction or short distance between adjacent nanoparticles, which gives rise to highly intense and localized electromagnetic fields. The embedded structure facilitates the nearby semiconductor to capture the strong field. The maximun control and utilization of this electromagnetic field significantly improve the formation of electron–hole pairs in the semiconductor photocatalysts, and ultimately enhance photocatalytic hydrogen evolution. The construction of coupled-metal nanostructures within the semiconductors represents an efficient way to enhance the solar energy conversion efficiency.

## Methods

All chemicals were analytic grade reagents and used without further purification. Purified water was used in all of the experiments.

**Preparation of Au colloids**. Au colloids were prepared by a sodium citrate reduction method. Typically, an aqueous solution of $HAuCl_4$ (0.25 mM, 100 mL) was heated to boiling, followed by the rapid addition of sodium citrate solution (0.5 M, 200 μL). The solution was kept boiling for another 15 min, producing a stable and deep-red dispersion of Au nanoparticles with an average diameter of about 15 nm (see in Supplementary Fig. 1). The citrate-protected Au-colloids suspension (denoted as Au-Cit) was then cooled to room temperature for next use.

**Preparation of Au nanochain embedded into $Zn_{0.67}Cd_{0.33}S$ (denoted as Au-chain@$Zn_{0.67}Cd_{0.33}S$)**. Au-chain@$Zn_{0.67}Cd_{0.33}S$ was prepared using a cysteine-assisted hydrothermal approach. Briefly, 100 mL L-cysteine solution (Cys, 60 mM) was mixed with $Zn(NO_3)_2$ and $Cd(NO_3)_2$ in a 1:0.5 molar ratio of Cys to $Zn^{2+}$/$Cd^{2+}$. The mixture was stirred for 30 min to form the stable complexes of cysteine-$Zn^{2+}$/$Cd^{2+}$. Then Au-Cit colloids (0.25 mM, 1 mL) was added to the freshly prepared cysteine-$Zn^{2+}$/$Cd^{2+}$ solution under vigorous stirring for 30 min, leading to a complete coupling between amine group of Cys and Au nanoparticles surface. Subsequently, the cysteine-$Zn^{2+}$/$Cd^{2+}$-coupled Au colloids were diluted to a total volume of 70 mL with deionized water and transferred into 100 mL Teflon-lined stainless-steel autoclaves. The autoclaves were maintained at 130 °C for 6 h and then cooled to room temperature naturally. The products were filtered and washed with distilled water to remove remaining ions and impurities. After that, the products were fully dried at 80 °C in an oven to obtain the final product. The amount of Au in the Au@$Zn_{0.67}Cd_{0.33}S$ composites was controlled by the amount of Au-Cit colloids in the synthesis process. The stoichiometric ratio is controlled at about 1.0 wt%. As a reference, pure $Zn_{0.67}Cd_{0.33}S$ solid solution was prepared with the same procedure as described above without adding Au-Cit colloids.

**Characterization methods**. TEM images were taken on JEM-2100F with an accelerating voltage of 200 kV equipped with an energy-dispersive spectroscopy analyzer. Powder X-ray diffraction (XRD) patterns were recorded on a Rigaku X-ray diffractometer using Cu Kα radiation ($\lambda = 1.5418$ Å). UV–vis diffused reflectance spectra of the samples were obtained from UV–vis–NIR spectrophotometer (Shimadzu-3600). XPS was performed on a Thermo ESCA LAB 250 system with MgKα source (1254.6 eV). The binding energies were calibrated using C 1s peak at 284.6 eV as standard. Raman spectra was measured at room temperature equipped with an Ar laser working at wavelengths of 532 nm (Lab-RAM HR Evolution, Horiba). The PL measurement was carried out on the FLS920 (Edinburgh Instrument) at room temperature using the excitation wavelength of 390 nm.

**FDTD simulation**. The near-field distributions of Au nanoparticles were simulated by the three-dimensional FDTD method. The TF/SF source was used in simulation. The incident wavelength is 520 nm. The grid spacing is 0.5 nm. The radius of gold nanosphere is 7.5 nm. The dielectric properties for gold are taken from ref. [44]. The background index is set to 1.0.

**Photocatalytic reaction**. The photocatalytic $H_2$ evolution reactions were carried out in a flowing gas diffluent system. The catalyst powder (0.1 g) was dispersed by a magnetic stirrer in 100 mL of 0.35 M $Na_2S$ and 0.25 M $Na_2SO_3$ aqueous solution in a reaction cell made of Pyrex glass. The reaction temperature was maintained at 15 °C. The reaction solution was evacuated 30 min to ensure complete air removal prior to light irradiation. Magnetic stirring was used to keep the photocatalyst particles in a suspension state. A 300 W Xe-lamp with a cutoff filter was employed for visible-light ($\lambda \geq 420$ nm) irradiation. The amounts of evolved $H_2$ was determined by an online gas chromatograph (GC 8 A, TCD) equipped with a 4 m 5 A molecular sieve columns and Ar as carrier gas.

The apparent quantum efficiency was measured under the same photocatalytic reaction except for the wavelength of irradiation light. The apparent quantum efficiency of different amounts of photocatalysts in one continuous reaction under visible light with different wavelengths of 420, 450, 500, 550, 600 nm were measured. Apparent quantum efficiency at different wavelengths was calculated by the following function. The band-pass and cutoff filters and a photodiode were used in measurement.

$$AQE\,(\%) = \frac{\text{Number of reacted electrons}}{\text{Total number of incident photons}} \times 100 \tag{1}$$

$$= \frac{2 \times \text{The number of evolved } H_2 \text{ molecules}}{\text{Total number of incident photons}} \times 100 \tag{2}$$

## Data availability

The data that support the findings of this study are available within the article and its Supplementary Information, and all data are available from the corresponding authors upon request.

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

## Acknowledgements

The authors acknowledge support from the National Natural Science Foundation of China (21473073, 11304164, 21972053), "13th Five-Year" science and technology research of the Education Department of Jilin Province (2016403), the Development Project of Science and Technology of Jilin Province (20170101171JC, 20180201068SF), the Open Project of State Key Laboratory of Inorganic Synthesis and Preparative Chemistry (201703) and the Fundamental Research Funds for the Central Universities.

## Author contributions

G.Y. and G.L. designed the experiment and performed the measurements and data analysis. J.Q. contributed to the theoretical simulation. B.Z. contributed to photocatalytic test. W.Z and W.Y. contributed to the optical measurement. G.Y., P.Z. and G.L. contributed to the discussion and the writing of the manuscript. All authors commented on the manuscript.

## Competing interests

The authors declare no competing interests.
