## [Peer Review File · Nature Communications]

Reviewers' comments:

Reviewer #1 (Remarks to the Author):

The authors reported their investigations on plasmon enhanced visible-light-driven hydrogen evolution. Basically they formed a gold-nanochain structure and embedded it into a semiconductor of Zn_{0.67}Cd_{0.33}S. They found that such a strategy led to high H₂ evolution activity and they attributed it to the coupling of Au plasmons, which led to an intensive near field that boosts the formation rate and lifetime of electron-hole pairs in the nearby semiconductor. They also compared the nanochain with the cases where individual Au nanoparticles were embedded or were placed on the surface. The work is interesting and manuscript is well written. But some important issues need to be addressed, which can help largely increase the impact of this manuscript.

1. It is good to design catalysts and obtain high activity. It would be even better if meanwhile new physical insights could be offered. For instance, in one control experiment, Au nanoparticles were anchored on the surface. In this case, the risk of the recombination of plasmon-induced electrons and holes could be reduced since they become immediately available for reactions. On this regard, isn't the sit-on (or close to)-surface design better? More discussion is needed on this aspect.

2. Related to the above comment, prof. Thomann's group published a well cited article "Plasmon enhanced solar-to-fuel energy conversion" in Nano letters 11 (8), 3440-3446, . In that work, they described and clearly explained why indeed the Au nanoparticles close to/on the surface behaved better than those embedded inside the semiconductor. I understood that in the current case, Au nanochains were formed. But the principle is expected to be the same. In addition, the authors also found that: " Au@Zn_{0.67}Cd_{0.33}S-S and Au@Zn_{0.67}Cd_{0.33}S-I enhance this activity to 8290 (molh⁻¹g⁻¹) and 11560 (molh⁻¹g⁻¹) ". Therefore, there are inconsistency. Please give comments.

3. Please better explain the novelty. The coupling to enhance near field and embedment in semiconductor to increase the use of the near field effect have been reported.

4. For comparison with embedded individual Au nanoparticles and those on the semiconductor surface, do they have the same size and size distribution as those in nano-chain samples? How about the Au concentration? Is it the same?

5. It has been reported that when plasmonic and semiconductor are too close, quenching can happen. Many cases have been reported; it was why plasmon nanostructures were usually coated by a thin shell to avoid the sinking effect when the energy transfer mechanism is expected to be dominant.

6. The authors mention: "It indicates that Au nanochain endows the photoexcited electrons with much longer lifetime." Plasmon-enhanced emission has been extensively studied. Emission enhancement in most of the cases is due to the increase of radiative recombination rate, which is accompanied by a shorter lifetime. Why a longer lifetime is claimed herein?

Reviewer #2 (Remarks to the Author):

The Authors present a nicely done study showing how different nanoparticle configurations lead to different changes in the lifetime, absorption, and photocatalytic activity of a ZnCdS catalyst. The plasmonic boost is similar to reported in literature, but using a higher efficiency ZnCdS catalyst as the starting point, instead of the routine TiO₂, leads to good performance figures. I think the paper is worth being published in nature communications if the following points can be met.

1. From the results, the lifetime and hydrogen generation are both increased in the order of clustered > embedded but not clustered > on the surface. As the author points out, this means one of the primary roles of the Au is to create a less-defective ZnCdS or some internal band bending. This is very interesting, and it makes the reader curious exactly what the optical function of the metal is. Could the authors please do the apparent quantum yield (Figure 6s) for each sample, including ZnCdS alone? From Figure 6a, there seems to be little resonant plasmon increase, but more just an increase in the overall absorption from scattering. The other samples would make it very clear how the plasmon versus the Au interface enhanced the photocatalysis when clustered, which is the main scientific finding.

2. How is the stability of the ZnCdS? For this process. If sacrificial agents are used, they should be clearly explained.

3. Please correct the use of surface plasmon resonance (SPR) versus localized surface plasmon resonance (LSPR).

4. The naming of the samples as -C, -I, and -S is hard to interpret. Please see if you can find a better convention for aiding readers.

Response to Reviewers

Reviewer #1:

The authors reported their investigations on plasmon enhanced visible-light-driven hydrogen evolution. Basically they formed a gold-nanochain structure and embedded it into a semiconductor of $Zn_{0.67}Cd_{0.33}S$. They found that such a strategy led to high H_2 evolution activity and they attributed it to the coupling of Au plasmons, which led to an intensive near field that boosts the formation rate and lifetime of electron-hole pairs in the nearby semiconductor. They also compared the nanochain with the cases where individual Au nanoparticles were embedded or were placed on the surface. The work is interesting and manuscript is well written. But some important issues need to be addressed, which can help largely increase the impact of this manuscript.

1. It is good to design catalysts and obtain high activity. It would be even better if meanwhile new physical insights could be offered. For instance, in one control experiment, Au nanoparticles were anchored on the surface. In this case, the risk of the recombination of plasmon-induced electrons and holes could be reduced since they become immediately available for reactions. On this regard, isn't the sit-on (or close to)-surface design better? More discussion is needed on this aspect.

Response: We agree with the point of the reviewer that, considering the recombination of electrons and holes, Au nanoparticles closer to the surface of the semiconductor are more efficient than the deeply embedded ones. However, based on the utilization of the electromagnetic field of plasmonic Au, all these Au nanoparticles

still should be completely embedded. Due to simply sitting on the surface of the semiconductor, only part of the electromagnetic field of Au nanoparticles can be used.

We provide three simple models to explain the points made above (see below). It is known that the localized electromagnetic fields of plasmonic metals are spatially non-homogenous. The highest intensity is at the surface of the plasmonic metal and decreases exponentially as the distance from the surface increases. Therefore, it is only in Model III that the semiconductor could fully take advantage of the localized electromagnetic fields of plasmonic Au.

Fig. R1 Illustration for different position of Au nanoparticles in plasmonic Au-semiconductor block.

A problem that needs to be solved in the future is the most suitable distance between the surface and the gold nanoparticle (see in Model III). It should be an optimized value that considers both the utilization of electromagnetic fields and recombination of electron-hole pairs. This distance should be different for the plasmonic block with different elemental compositions, since the plasmonic metal determines the strength of the electromagnetic field and the semiconductor affects the carrier diffusion length. It needs a series of detailed experiments to prove. We believe that, in future work, precise control of the depth of embedded Au particles will further

optimize the performance of the catalysts.

According to the suggestion of the reviewer, the above discussion on the position of Au nanoparticles in plasmonic Au-semiconductor block was added in the revision (see in line 23 on page 17 to line 7 on page 18).

2. Related to the above comment, Prof. Thomann's group published a well cited article "Plasmon enhanced solar-to-fuel energy conversion" in Nano letters 11 (8), 3440-3446, . In that work, they described and clearly explained why indeed the Au nanoparticles close to/on the surface behaved better than those embedded inside the semiconductor. I understood that in the current case, Au nanochains were formed. But the principle is expected to be the same. In addition, the authors also found that: "Au@Zn_{0.67}Cd_{0.33}S-S and Au@Zn_{0.67}Cd_{0.33}S-I enhance this activity to 8290 $\mu\text{molh}^{-1}\text{g}^{-1}$ and 11560 $\mu\text{molh}^{-1}\text{g}^{-1}$ ". Therefore, there is inconsistency. Please give comments.

Response: We appreciate the reviewer for recommending this article to discuss our results. The systematic theoretical simulations of Prof. Thomann's work are very helpful for us. Their theoretical results showed that the strength of the electromagnetic field is quite similar, whether the Au nanoparticles are on the top or the bottom of the Fe₂O₃ layer (Figure 2 and 3 of Thomann's report). It could help us exclude the influence from the differing of the electromagnetic field caused by the placement of Au nanoparticles.

We noticed that there is some difference in their experimental results in comparison

with our work. The enhancement of the measured photocurrent for Au-top is higher than that for Au-bottom. The authors ascribed most of the enhancement to the property changes of iron oxide or introducing new catalytic effects (see in Thomann's report, line 52 to 58, right column of page 3442).

Moreover, in our opinion, there are at least three different factors between Prof. Thomann's report and our work. All these three factors could significantly affect the change of photocatalytic activity. One is the poor electronic transport property of Fe_2O_3 versus the excellent carrier diffusion property of sulfide compounds, including $\text{Zn}_{0.67}\text{Cd}_{0.33}\text{S}$. The second one is the size effects of Au nanoparticles. The dominant mechanisms for 50 nm Au (Prof. Thomann's report) are scattering mechanisms, whereas 15 nm Au (our work) are near-field electromagnetic mechanisms. The third one is the different catalytic system: photoelectric catalysis (Prof. Thomann's report) and powder photocatalysis (our work). In photoelectric catalysis, the influence of extra applied potential and interaction of semiconductor/FTO or ITO glass should also be considered.

Both the properties of the semiconductor and the size of Au nanoparticles support us to develop the embedding strategy. It could fully take advantage of energy from plasmonic Au, leading to a higher enhancement of Au-iso@ $\text{Zn}_{0.67}\text{Cd}_{0.33}\text{S}$ ($11560 \mu\text{molh}^{-1}\text{g}^{-1}$, embedding isolated Au nanoparticles) than Au-surf@ $\text{Zn}_{0.67}\text{Cd}_{0.33}\text{S}$ ($8290 \mu\text{molh}^{-1}\text{g}^{-1}$, Au nanoparticles loaded on the surface). In the revision, further discussion, combined with Prof. Thomann's theoretical simulations, is added (see in line 7 to 11, page 17). This report is also cited in the revision (reference 43).

3. Please better explain the novelty. The coupling to enhance near field and embedment in semiconductor to increase the use of the near field effect have been reported.

Response: The novelty of this work is further clarified in the revision. The main novelty of this work is building a strongly plasmon-coupled metal nanostructure into a semiconductor and its contribution to photocatalytic performance. As shown in the title of this work, collective excitation of plasmon-coupled Au-nanochain could boost photocatalytic hydrogen evolution of semiconductor. Based on this investigation, we expect to introduce a novel strategy to improve the efficiency of semiconductor photocatalysts for solar fuel production.

We know that the topics of coupled metallic nanostructures enhancing electromagnetic fields are hot ones in physics and nanoscience areas. Wrapping metal nanoparticles with a semiconductor is also widely studied in material science, such as various kinds of core-shell materials. However, the photocatalytic efficiency enhanced by current wrapping structures is still limited, and fabricating coupled metallic nanostructures into semiconductor still face great challenges. Therefore, few works were reported that focused on the effect of coupled metallic nanostructures on photocatalytic performance.

Our case proves the significant contribution of collective behavior of plasmonic metals from experiment and theoretical aspects. Considering a great deal of coupled metallic nanostructures has been reported in physics and nanoscience areas,

embedding such nanostructures into semiconductors could further optimize the efficiency of photocatalysts. As such, we think our work would open a promising way for photocatalyst development. In the revision, more explanation of novelty is added, mainly located at the introduction and discussion section (see in line 17 to 21 on page 5 and line 19 to 22 on page 17).

4. For comparison with embedded individual Au nanoparticles and those on the semiconductor surface, do they have the same size and size distribution as those in nano-chain samples? How about the Au concentration? Is it the same?

Response: Yes, both the size and concentration of Au nanoparticles were well controlled in these three samples. Au sources were all from the same batch of Au colloid nanoparticles. They are prepared by a sodium citrate reduction method. This method is very easy and reproducible. TEM images shows the particle could be controlled at 15 ± 2 nm (Supplementary Figure 1). The concentration of Au nanoparticles in these three samples is controlled at about 1.0 wt% (described in the Method section). We also verified the reproducibility of these $Zn_{0.67}Cd_{0.33}S$ containing Au samples when we prepared the first draft of this article. The results show that they are all reproducible under our experimental conditions. These results can ensure the comparison is meaningful.

5. It has been reported that when plasmonic and semiconductor are too close, quenching can happen. Many cases have been reported; it was why plasmon

nanostructures were usually coated by a thin shell to avoid the sinking effect when the energy transfer mechanism is expected to be dominant.

Response: We agree with the viewpoint of the reviewer. We noticed this point when we designed the catalysts. In our case, Au nanoparticles are protected with citric acid group to homogeneously disperse in Au colloids aqueous solution. During the hydrothermal synthesis, *L*-cysteine would couple with these Au nanoparticles. It can be proposed that a non-conductive organic layer would form during these two processes, which could effectively avoid the sinking effect. We can get some evidence from the FT-IR spectrum, which shows vibration signals assigned to organic groups such as $-\text{COO}^-$. We thank the reviewer for the suggestion. Corresponding discussion is added in the revision (line 19 on page 16 to line 2 on page 17, and Supplementary Figure 17).

Fig. R2 FTIR spectra of *L*-cysteine and Au-chain@Zn_{0.67}Cd_{0.33}S samples.

6. The authors mention: “It indicates that Au nanochain endows the photoexcited electrons with much longer lifetime.” Plasmon-enhanced emission has been extensively studied. Emission enhancement in most of the cases is due to the increase

of radiative recombination rate, which is accompanied by a shorter lifetime. Why a longer lifetime is claimed herein?

Response: We understand the reviewer's concerns. In many regular cases, emission enhancement is accompanied by a shorter lifetime due to the increase of radiative recombination rate ($1/\tau = \gamma_{\text{rad}} + \gamma_{\text{non-rad}}$), however, we also observed trends in literature similar to our work. We think one possible reason is the significant enhancement of electron-hole pairs by introducing plasmonic Au. It is known that time-resolved photoemission spectroscopy is to probe the photo-products upon the time delay. The total amount of electron-hole pairs is significantly increased by the electromagnetic field of plasmonic Au. Such large amount of charge carriers could lead to the long lifetime signal observed upon the time delay.

In addition, the special surface properties of $\text{Zn}_{0.67}\text{Cd}_{0.33}\text{S}$ should be an important factor responsible for the long lifetime signal. A large amount of low valent Cd (mainly Cd^+) is present on the surface of the $\text{Zn}_{0.67}\text{Cd}_{0.33}\text{S}$ semiconductor. It can be identified by fitting the XPS spectra of these samples. This point was investigated in great detail in our previous work (*J. Mater. Chem. A*, 2016, 4, 13803 and *Chem. Commun.*, 2015, 51, 10676). These low valent Cd species formed essentially due to the surface S vacancies, and effectively act as trap sites of photo-excited electrons. These trap sites induce Stokes-shifted emissions (Fig.3B, 470-650 nm). In our current work, the significant enhancement of emissions is located at this area (Fig. 4c). The increased photo-excited electrons could be trapped by these sites. These deeply trapped sites could extend their lifetime. More discussion is added in the revision based on

above analysis (line 7 to 13 on page 16).

Fig. R3 Cd 3d XPS spectra of pure Zn_{0.67}Cd_{0.33}S and Au@Zn_{0.67}Cd_{0.33}S with different spatial arrangement of Au nanoparticles.

Reviewer #2 (Remarks to the Author):

The Authors present a nicely done study showing how different nanoparticle configurations lead to different changes in the lifetime, absorption, and photocatalytic activity of a ZnCdS catalyst. The plasmonic boost is similar to reported in literature, but using a higher efficiency ZnCdS catalyst as the starting point, instead of the routine TiO₂, leads to good performance figures. I think the paper is worth being published in nature communications if the following points can be met.

1. From the results, the lifetime and hydrogen generation are both increased in the order of clustered > embedded but not clustered > on the surface. As the author points out, this means one of the primary roles of the Au is to create a less-defective ZnCdS or some internal band bending. This is very interesting, and it makes the reader

curious exactly what the optical function of the metal is. Could the authors please do the apparent quantum yield (Figure 6s) for each sample, including ZnCdS alone? From Figure 6a, there seems to be little resonant plasmon increase, but more just an increase in the overall absorption from scattering. The other samples would make it very clear how the plasmon versus the Au interface enhanced the photocatalysis when clustered, which is the main scientific finding.

Response: We agree with the comments of the reviewer. Our work shows that loading the plasmonic Au on the surface of the semiconductor could improve the activity in a certain degree, but the effect is much lower than that of samples with embedded structures. According to the suggestion of the reviewer, the apparent quantum yields of $\text{Zn}_{0.67}\text{Cd}_{0.33}\text{S}$ and all the Au-containing samples are detected in the revision. As shown in Supplementary Figure 6, the apparent quantum yield at 420 nm is 14.2%, 19.3%, 28.7% and 43.5% for these four samples, respectively. It should be noted that all these investigations are carried out under similar reaction conditions as those for H_2 evolution rate tests. It is known that the H_2 evolution rate is 5020, 8290, 11560 and 16420 $\mu\text{molh}^{-1}\text{g}^{-1}$ over these four samples. The enhancements of these apparent quantum yields are consistent with the H_2 evolution rate. The apparent quantum yield could be further optimized. When the amount of catalyst is increased to 0.3 g, the apparent quantum yield could be optimized to 54.6%. It is mainly due to the full use of light intake when the amount of catalyst increased. These results are added in the revision (line 10 to line 14 on page 14, Supplementary Figure 6).

Fig. R4 Apparent quantum yield of pure Zn_{0.67}Cd_{0.33}S and different Au-containing samples under different wavelength illumination.

As for the optical properties, the shift is an interesting couple effect of plasmonic metals (*Langmuir*, 2012, 28, 8862 and *Chem. Rev.* 2011, 111, 3913). In our work, we proposed that the change of optical properties of Zn_{0.67}Cd_{0.33}S are affected by the electromagnetic effect of the coupled Au nanostructure. The main finding of our work is the contribution of collective behavior from plasmonic Au nanoparticles, i.e. the coupled effect, to the semiconductor. We believe that it would become a novel and effective strategy to improve the efficiency of photocatalysts. In the revision, more explanation of novelty is added, mainly located at the introduction and discussion section (see in line 17 to 21 on page 5 and line 19 to 22 on page 17).

2. How is the stability of the ZnCdS? For this process, if sacrificial agents are used, they should be clearly explained.

Response: We agree with the reviewer that stability is an important parameter of the photocatalysts, especially the sulfide compounds. In the revision, both the stability of Zn_{0.67}Cd_{0.33}S and Au-nanochain embedded sample are detected. As shown in Fig. R5

(Supplementary Figure 13), the H₂ evolution rate of Au-nanochain embedded sample has almost no change after more than 40 h detection. As for pure Zn_{0.67}Cd_{0.33}S, a decreasing trend can be observed when the reaction time lasts more than 25 h. This result shows that Zn_{0.67}Cd_{0.33}S in the Au-nanochain embedded sample is quite stable under the reaction conditions. Corresponding Figure and description are added in the revision (line 16 to line 17 on page 14, Supplementary Figure 13).

Fig. R5 Visible-light-driven H₂ evolution rate as a function of time. Reaction condition: 0.1 g photocatalysts in 100 mL Na₂S (0.35 M)-Na₂SO₃ (0.25 M) solution, 300 W Xe-lamp equipped with cut-off filter ($\lambda \geq 420$ nm).

3. Please correct the use of surface plasmon resonance (SPR) versus localized surface plasmon resonance (LSPR).

Response: The description has been corrected in the revision. We thank the reviewer for the correction.

4. The naming of the samples as -C, -I, and -S is hard to interpret. Please see if you can find a better convention for aiding readers.

Response: The Au-containing samples are renamed in the revision. They are Au-chain@Zn_{0.67}Cd_{0.33}S, Au-iso@Zn_{0.67}Cd_{0.33}S and Au-surf@Zn_{0.67}Cd_{0.33}S, respectively. We think these names would be easier to understand by readers. We thank the reviewer for pointing them out.

REVIEWERS' COMMENTS:

Reviewer #1 (Remarks to the Author):

I recommend its publication because the revised version appropriately addressed most of my comments. But I would suggest the authors further explain in the manuscript clearly how come the longer lifetime can be due to the increase number of electron-hole pairs? On the other hand, if it is due to increase of trapped sites as the authors mentioned, the emission wavelength should be different.

Reviewer #2 (Remarks to the Author):

Based on the revisions, it is clear that somehow the coupled metal nanoparticles are leading to an increase in lifetime that is stronger than the isotropically distributed metal nanoparticles. The apparent quantum yield and lifetime suggest the enhancement is not a plasmon mediated one, but instead a decrease in the carrier recombination. Based on the absorption spectrum, there may be some distributed field effect or scattering when the particles are close together. Whether there is a link between the local field and lifetime, or it is just a morphological change, can not be pinned down.

As long as these points are clear in the final manuscript, and not just that the performance is from some non-descriptive plasmonic effect, I think this work can be published. The enhancement of semiconductor photocatalysis with plasmonics is not new, but the mechanism at work here, whatever it is, is sufficiently different to spark interest.

Response to Reviewers

Reviewer #1 (Remarks to the Author):

I recommend its publication because the revised version appropriately addressed most of my comments. But I would suggest the authors further explain in the manuscript clearly how come the longer lifetime can be due to the increase number of electron-hole pairs? On the other hand, if it is due to increase of trapped sites as the authors mentioned, the emission wavelength should be different.

Response: I agree with the reviewer that lifetime of photo-excited electron-hole pairs is an important factor in photocatalysis. In the revision, further explanation on the longer lifetime of PL decay spectra is added. More analysis based on the increasing number of electron-hole pairs and the presence of surface trapped states is provided (see in page 15 and 16, highlighted section).

As for the emission wavelength, no obvious shift among these samples should be mainly due to the similar crystal structure and surface properties of $\text{Zn}_{0.67}\text{Cd}_{0.33}\text{S}$ semiconductor in these samples. The difference is that these similar trapped sites play a more important role in Au-containing samples with the increase number of electron-hole pairs. In many literatures, the shift of the emission wavelength usually results from the different physicalchemical properties of the samples. For example, Durrant *et al.* observed a red-shift emission over $g\text{-C}_3\text{N}_4$ prepared at different temperatures (*J. Am. Chem. Soc.*, 2017, 139, 5216). It is due to the formation of different trapped states under corresponding temperatures. About this point, we also provide explanation in the revision (see in line 23 on page 15 to line 2 on page 16).

Reviewer #2 (Remarks to the Author):

Based on the revisions, it is clear that somehow the coupled metal nanoparticles are leading to an increase in lifetime that is stronger than the isotropically distributed metal nanoparticles. The apparent quantum yield and lifetime suggest the enhancement is not a plasmon mediated one, but instead a decrease in the carrier recombination. Based on the absorption spectrum, there may be some distributed field

effect or scattering when the particles are close together. Whether there is a link between the local field and lifetime, or it is just a morphological change, can not be pinned down.

As long as these points are clear in the final manuscript, and not just that the performance is from some non-descriptive plasmonic effect, I think this work can be published. The enhancement of semiconductor photocatalysis with plasmonics is not new, but the mechanism at work here, whatever it is, is sufficiently different to spark interest.

Response: I agree with the point of the reviewer that the coupled metal nanoparticles lead to an increase in lifetime of photo-excited electron-hole pairs. The significant enhancement of photocatalytic activity should originate from the collective behavior of these coupled metal nanoparticles, while not simple plasmonic effect of the isolated one. The result of this behavior is a decrease in the carrier recombination. These points were further clarified in the revision (see in page 17, highlighted section). Thank the reviewer for the suggestion and the analysis.